# Multiscale Mathematical Modeling in Systems Biology: A Framework to Boost Plant Synthetic Biology

**DOI:** 10.3390/plants14030470

**Published:** 2025-02-05

**Authors:** Abel Lucido, Oriol Basallo, Alberto Marin-Sanguino, Abderrahmane Eleiwa, Emilce Soledad Martinez, Ester Vilaprinyo, Albert Sorribas, Rui Alves

**Affiliations:** 1Systems Biology Group, Department Ciències Mèdiques Bàsiques, Faculty of Medicine, Universitat de Lleida, 25008 Lleida, Spain; abel.lucido@udl.cat (A.L.); oriol.basallo@udl.cat (O.B.); alberto.marin@udl.cat (A.M.-S.); abdrahman.eleiwa@gmail.com (A.E.); em19@alumnes.udl.cat (E.S.M.); ester.vilaprinyo@udl.cat (E.V.); albert.sorribas@udl.cat (A.S.); 2Institut de Recerca Biomèdica IRBLleida, 25198 Lleida, Spain; 3MathSys2Bio, Grup de Recerca Consolidat de la Generalitat de Catalunya, 25001 Lleida, Spain; 4National Institute of Agricultural Technology (INTA), Pergamino 2700, Argentina

**Keywords:** plant science, multiscale, synthetic biology, mathematical model

## Abstract

Global food insecurity and environmental degradation highlight the urgent need for more sustainable agricultural solutions. Plant synthetic biology emerges as a promising yet risky avenue to develop such solutions. While synthetic biology offers the potential for enhanced crop traits, it also entails risks of extensive environmental damage. This review highlights the complexities and risks associated with plant synthetic biology, while presenting the potential of multiscale mathematical modeling to assess and mitigate those risks effectively. Despite its potential, applying multiscale mathematical models in plants remains underutilized. Here, we advocate for integrating technological advancements in agricultural data analysis to develop a comprehensive understanding of crops across biological scales. By reviewing common modeling approaches and methodologies applicable to plants, the paper establishes a foundation for creating and utilizing integrated multiscale mathematical models. Through modeling techniques such as parameter estimation, bifurcation analysis, and sensitivity analysis, researchers can identify mutational targets and anticipate pleiotropic effects, thereby enhancing the safety of genetically engineered species. To demonstrate the potential of this approach, ongoing efforts are highlighted to develop an integrated multiscale mathematical model for maize (*Zea mays* L.), engineered through synthetic biology to enhance resilience against *Striga* (*Striga* spp.) and drought.

## 1. Introduction

The first green revolution successfully addressed the food crisis of the 1960s, by intensifying crop production under favorable environmental conditions with the aid of excessive fertilizers [1]. This approach led to an increase in the environmental impact of agriculture that will be charged to the current human generation [1]. These costs include land degradation due to depleted soil nutrients, water eutrophication from fertilizer residues, the over-extraction of groundwater, pesticide overuse leading to pest resistance, the extinction of indigenous crop varieties, and increased greenhouse gas emissions contributing to climate change [1,2,3]. Climate change has intensified extreme weather conditions, resulting in severe flooding and droughts that jeopardize crop production in Asia [4]. In sub-Saharan Africa, *Striga hermonthica* remains the most destructive pest, causing up to 100% yield loss in some crop fields [5]. In addition, the escalating climate crisis, ongoing conflicts, and recurring pest outbreaks are collectively undermining global crop production. For example, the conflict between Russia and Ukraine is jeopardizing global maize production, as the two countries have a combined global production share of 17% for that cereal [6,7]. The continuous increase in world population also raises concerns about global food insecurity [8]. Projections by the Organization for Economic Co-operation and Development (OECD) and the Food and Agriculture Organization (FAO) of the United Nations (UN) estimate a world population of nearly 10 billion by 2050. This scenario is raising the stakes for sustainable food production and there are calls for a second Green Revolution that emphasizes the production of crops that are resilient to suboptimal environmental conditions and create a more manageable environmental footprint [9].

To meet these challenges, researchers are turning to synthetic biology in plants, which employs biological engineering principles to improve beneficial plant traits in a way that goes beyond the speed and capacity of natural processes [10]. Through metabolic engineering and synthetic biology, plant scientists could remodel metabolic pathways to increase the production of useful metabolites or inhibit the synthesis of less useful ones [11]. Synthetic biology products now available in the market include Impossible Foods’ bleeding plant-based burgers, created from an engineered strain of the yeast (*Pichia pastoris*) that produces soy leghemoglobin and gives the burger a meaty sensation [12,13]. There is also Zymergen’s polyimide film, named hyaline, that is made from bio-sourced monomers [14], Pivot Bio’s biological nitrogen fertilizer for cereals [15], and Calyxt’s high-oleic oil from soybeans called Calyno [16,17]. More recently, printed vegan salmon also became available in supermarkets and many vegan fish alternatives are currently being developed [18]. As such, plant synthetic biology holds immense potential to steer agriculture towards a more sustainable future, alleviating global food insecurity.

While redesigning crops through plant synthetic biology is undeniably promising for the future of agriculture, it can also cause environmental damage. One of the risks that comes together with synthetic biology is producing a genetically engineered species that may increase ecosystemic problems once released into the natural environment [19,20]. Engineering plant genes may also initialize pleiotropic effects to the organism, due to off-target effects that can lead to unforeseen changes in the target species [19]. These risks can be mitigated using the combination of plant synthetic biology and mathematical modeling, which allows the identification of mutational targets using parameter estimation, bifurcation analysis, sensitivity analysis, and optimization [21,22].

Aside from biological and environmental risks, synthetic biology also presents economic and ethical challenges. Economically, a study on the economics of synthetic biology [23] highlighted the potential “winner-take-all” scenario, which could lead to the monopolization of a company. Still, preserving the balance between open-source platforms and patent protections offers a promising approach to managing this risk. Ethically, challenges include ensuring fair and intergenerational accessibility and maintaining public trust. Existing ethical governance frameworks [20,24,25] provide guidelines for addressing these issues. Concerning in silico investigations, the development of open-source, user-friendly platforms capable of predictive simulation could play a key role in communicating the risks and benefits of synthetic biology to the public.

To improve risk mitigation, integrated mathematical models should be applicable at different levels and with varying granularity. This is because plants are complex systems composed of roughly modular components, and this modularity extends from the genome level to the tissue and organ levels [26]. However, the application of multiscale mathematical models, while frequent in microbial systems [27,28,29,30], remains underutilized in plants. Integrating technological advancement in agricultural data analysis facilitates a comprehensive understanding of crops, from genotype to phenotype to ecosystems, promoting effective environmental conservation and enhanced agricultural productivity [31,32].

To advance the creation and use of multilevel modeling in plants, this study reviewed common modeling approaches and methodologies applicable to plants at various scales, from molecular to crop levels. This review emphasized the most commonly used approaches, ranging from the simplest to the more mathematically complex methods. Alternatives for integrating models across scales were also explored, with a discussion of the advantages and shortcomings of each approach. Finally, the review provided insights into ongoing efforts to develop an integrated multiscale mathematical model for maize, engineered through synthetic biology to enhance resilience against *Striga* and droughts.

## 2. Model Building

Building multiscale mathematical models in plant synthetic biology remains a challenge, and as the model becomes more complex, so does its mathematical analysis. Further, integrating models of different levels and granularities poses additional complex challenges. In response to these challenges, the review outlined an iterative workflow for constructing a multiscale mathematical model in plant synthetic biology.

Figure 1 illustrates two nested cycles that summarize the workflow. This portrays a minimalistic methodology for mathematical modeling that spans multiple scales, from the molecular to the plant level. The inner cycle addresses the mathematical modeling methodology subgrouped into input, method/approach, and output, while the outer cycle depicts the integration of processes across different scales of the plant system. The inner workflow is applied at each step of the outer cycle to construct a multiscale mathematical model of a plant. Once validated, this model can be employed to predict unintended pleiotropic effects of manipulating the genome on the plant at various levels.

The remainder of this section reviews modeling strategies employed in plants at various scales, from molecular to plant level. The review is framed from the perspective of the workflow summarized in Figure 1. The stages proposed in Figure 1 may expand to include more or less detailed modeling of the various plant subsystems. For instance, shoot architecture can be modeled from a coarse level or at a more detailed level, where branching, leaves, and flowers are explicitly considered.

### 2.1. Metabolic Modeling with FBA, Kinetic, or Hybrid Approaches

#### 2.1.1. Applications

Metabolic modeling techniques, such as flux balance analysis (FBA), kinetic modeling, and hybrid approaches, play a crucial role in driving metabolic engineering towards efficiency [33]. Within the iterative design–build–test–learn (DBTL) cycle of metabolic engineering, metabolic modeling emerges as a valuable tool [11,34]. It offers a unique perspective on plant metabolism, yielding quantitative insights and informing new engineering strategies [34]. Particularly, it streamlines the trial-and-error cycles inherent in metabolic engineering, saving considerable time and resources [11,35,36].

The initial steps involve the collection of pathway data in order to build a metabolic model and represent the pathway mathematically for analysis. When pathways are unavailable in databases, reconstruction and inference based on literature or genome analysis become necessary. This process may involve the use of available gene sequences to identify related pathways and their associated data. Once the relevant pathways are defined, a stoichiometric matrix can be derived and utilized for model construction. Additionally, preexisting kinetic information is often required for model creation and analysis. Appendix A provides examples of databases containing kinetic and pathway data that are available to the reader for reference.

Most studies modeling plant metabolism employ flux balance analysis (FBA)-based or kinetic-based models [34,36]. Flux balance analysis (FBA) is a widely used constraint-based approach for analyzing material and energy flow within metabolic networks [37,38]. Its advantage lies in the ability to analyze entire metabolic networks without detailed process mechanism information. FBA models have been extensively used to study plant metabolism, including in species such as thale cress *(Arabidopsis thaliana* L.), tomato (*Solanum lycopersicum* L.), maize, and rice (*Oryza sativa* L.) [38,39,40,41,42,43,44,45,46,47,48,49]. They facilitate understanding flux distributions and gene essentiality, requiring modest prior information.

However, FBA has limitations. It cannot accurately represent regulation and lacks information on transient responses or metabolite concentrations, which are crucial in metabolic engineering. For such cases, more detailed kinetic models are necessary, either developed from scratch or based on simplified FBA versions. Kinetic models offer mechanistic descriptions of metabolism, enabling regulation incorporation but demanding more prior information. They require knowledge of individual metabolic processes and parameter values for rate expressions, often obtained from databases like BRENDA, MetaCyc, or KEGG (Appendix A). When using these databases, it is important to be aware that parameter values are frequently determined in vitro, and might be significantly different from in vivo values [36].

As FBA models, kinetic models can be used to analyze steady-state flux distributions. In addition, they provide information about transient flux changes and concentrations, both at steady state and during transient responses. This is required in situations where metabolic engineers need to predict the behavior of the biological system under genetic or environmental perturbations [36]. In addition, kinetic modeling enables sensitivity and stability analysis, which provide a framework for elucidating parameters responsible for controlling metabolic fluxes in a network [36,50,51,52].

Kinetic models typically focus on smaller parts of the metabolic network [34], employing rational expressions like Michaelis–Menten or Hill equations [53,54,55,56,57,58,59,60]. Other typical kinetic expressions include mass-action and Arrhenius equations [61], and saturable and cooperative formalisms [52,62,63], as well as generalized mass action (GMA) kinetics [51,64]. GMA and other approximate representations are highly useful when the mechanisms of individual processes are unknown [62,63]. Kinetic models were used to study, among other things, the flavonoid biosynthesis pathway in thale cress [61] and tomato [57], central carbon metabolism in potato (*Solanum tuberosum* L.) [58], central carbohydrate metabolism [56], aspartate-derived amino-acid pathways [54], and the 2-C-methyl-D-erythritol 4-phosphate (MEP) pathway [59] in thale cress, the benzenoid network in *Petunia hybrida* Vilm. [55], lignin biosynthesis in *Brachypodium distachyon* (L.) [64], and nitrate reductase activity [53].

FBA and kinetic models can sometimes be combined into hybrid models [36]. By integrating both methodologies, hybrid models overcome limitations and enhance model power. Creating a hybrid model is a two-step process that involves creating an FBA model to generate flux distributions, and then using these distributions to estimate parameter values for kinetic sub-models. Hybrid models provide detailed concentration insights from kinetic modeling while validating results with FBA outputs, offering a more accurate representation of real-world metabolic processes [65]. Another parameter determination approach is ensemble modeling which calculates the parameter values with references to the steady state fluxes [34,66].

#### 2.1.2. Creating a Metabolic Model

Figure 2 summarizes a typical workflow for metabolic modeling with FBA, kinetic, or hybrid approaches. The figure illustrates the input, method/approach, and output of a plant metabolic model. The modeling process is examined in greater detail below.

In general, given a metabolic pathway with m number of metabolites $\left(M_{i}\right)$ and n number of reactions/fluxes $\left(v_{j}\right)$, an m by n stoichiometric matrix (S) can be generated [67,68,69,70]. After obtaining S, a flux vector (v) with n by 1 dimension is created, where each row is the flux $\left(v_{i}\right)$. Matrix multiplication is then applied to S and v to get the metabolite/substrate vector (M) with m by 1 dimension where each row represents the rate of changes of each $\left(M_{i}\right)$. This relationship is expressed mathematically in Equation (1).(1)$$\frac{\mathrm{dM}}{\mathrm{dt}}=Sv$$

From Equation (1), the process diverges into different paths depending on the modeling approach. For using an FBA model, a steady state of flux is assumed, where the rate of change in the metabolite vector (M) is zero, that is $\frac{d\;M}{\mathrm{dt}}=0$. Hence, Equation (1) becomes(2)Sv=0.

At this stage, constraints are applied to the flux vector (v) using an objective function. This objective function may depend on the rate at which material/energy enters and exits the system, as well as the minimum and maximum amounts of flux that must flow through various sub-processes for the system to remain viable. Linear optimization techniques are then used to solve the flux distribution and find the appropriate solutions to the optimization problem.

To construct a kinetic model, the process begins with the mathematical representation of change in metabolite concentrations over time, as shown in Equation (1). Then, a suitable mathematical representation is selected to describe the kinetics of individual processes and their dependence on substances directly influencing flux. Various representations, including linear, rational, and power laws, among others, are available [63]. Each representation is associated with parameters whose values need to be estimated to predict the dynamic behavior of the model quantitatively. Initial parameter estimates can be obtained from databases, such as those listed in Appendix A, or from primary literature sources.

Equation (1) is equivalent to $\frac{\mathrm{dM}}{\mathrm{dt}}={\left[\frac{dM_{1}}{dt},\;\frac{dM_{2}}{dt},\;\ldots ,\;\frac{dM_{m}}{dt}\right]}^{T}$ which is a nonlinear ODE. In analyzing nonlinear ODEs, a common practice is to approximate using a linearization of the system. To linearize Equation (1), the Jacobian matrix (J) is calculated by getting the partial derivative of each ODE with respect to each dependent variable.

Using the Jacobian matrix (J), the stability of the system is determined through the computation of the eigenvalues of the matrix J. The real part of the eigenvalue of a stable system must be negative for the steady state to be stable. Bringing this into consideration, different methods are used to investigate how specific factors influence metabolite concentrations. For example, sensitivity analysis [68,69] identifies which parameters can cause significant changes, parameter scanning evaluates the extent of these changes, and time-series analysis tracks how metabolite concentrations change over time [51,60].

From Equation (2), the output of FBA yields a flux distribution *v* that minimizes or maximizes the given objective function. Common applications of these models involve modifying constraints and initial conditions or altering the system by removing selected processes from the diagram, thus assessing their impact on the model’s solution space. On the other hand, the output of the kinetic model encompasses all metabolic concentrations involved in the metabolic pathway, offering a comprehensive view of the system’s behavior under specified conditions.

### 2.2. Shoot and Root System Architecture

Shoot and root plant system architectures are essential for resource acquisition during plant development. Yet, conducting plant experiments, particularly in the field of plant synthetic biology, is inherently challenging with respect to time and cost. It is also hard to estimate the risk of disrupting local ecologies when introducing new plant varieties. Developing a 3D architecture visualization model is crucial to predicting plant shoot/root growth and development, facilitating reasonable approximate estimations of crop productivity [71,72]. Thus, there has been a growing interest in building mathematical models of the shoot and root system architecture to conduct in silico investigation in plants; some of these modeling efforts are listed in Appendix A. However, this field is still less mature than metabolic modeling and methodological development is ongoing.

#### 2.2.1. Shoot System Architecture

##### Applications

The shoot system architecture (SSA) is generally the above-ground portion of the plant that consists of stem, leaves, buds, flowers, and fruits. Optimizing the shoot growth of wheat (*Triticum aestivum* L.) revolutionized agriculture during the 1960s, resulting in the First Green Revolution. First, the agronomist Norman Borlaug developed a high-yield and disease-resistant wheat variety [73,74]. However, the wheat stalk of this variety was too thin to support the weight of its grain, leading to a type of breaking called lodging. Thus, he further developed a dwarf variety of this wheat which can bear heavier weight [74]. This development of dwarf wheat solved an impending global food crisis during that time.

Creating dwarf wheat through traditional breeding took many years [75]. With the current gene editing technology, it is already possible to speed up the process of developing more desirable plant varieties, creating specific modifications in the genome of plants to obtain specific traits. Using models to simulate the effects of these genetic modifications on the phenotype of SSA can help mitigate risks and predict the quantitative impact of genetic interventions on shoot traits and crop yield more accurately. This could speed up the process of producing new crop varieties.

There are several models for SSA that have been developed over the years [76,77,78,79]. L-systems underlie many of these models [76,79,80,81,82,83,84,85,86,87,88,89,90]. This modeling approach was named after Aristid Lindenmayer, and it is widely used to model plant morphogenesis [91]. In Appendix A, we listed several functional–structural plant modeling (FSPM) platforms from the Quantitative plant website [92]. These platforms can also be utilized to model root and whole plant architectures.

##### Creating a Shoot System Architecture

Figure 3 presents a workflow for building an SSA model. The process begins by determining whether the plant species is monocot or dicot, as they have different architecture from one another. A monocot generally has tillers, with secondary shoots branching from its primary shoot. A dicot typically has only one primary shoot [93].

Modeling SSA involves describing phytomers, which are the fundamental structural units of the shoot. These phytomers are produced from shoot apical meristems (SAMs), also called terminal buds [77,79,94]. Figure 3 shows an example of phytomers. Each phytomer is a stem segment consisting of a leaf, an axillary bud, a terminal bud, a node, and an internode. The phytomers are stacked up to form a shoot, whose form may vary depending on the plant species. Wen et al. [79] defined a geometrical model of a shoot as(3)$$Shoot_{i}^{n}=\overset{n}{\underset{j=1}{\cup }}R_{ij}\cdot T_{ij}\cdot Phytomer_{ij}.$$

Here, $R_{ij}$ and $T_{ij}$ describes the rotation and translation of the $Phytomer_{ij}$ and the $Shoot_{i}^{n}$ is the union of n number of phytomers.

These models can be used to quantify shoot architecture traits such as leaf area (an important factor for capturing light), plant height, stem length, volume, etc.

#### 2.2.2. Root System Architecture

##### Applications

The most frequently used root phenotyping techniques include shovelomics, the monolith method, and X-ray computer tomography [95,96,97]. Shovelomics [95] is a rapid and high-throughput phenotyping method. This is done by digging a shovel-sized portion of the root and then manually measuring 10 root architectural traits. The monolith method [96] uses a steel cylinder with 20 cm diameter and 30 cm depth for excavating roots and measuring their phenotypes. This method allows the excavation of deeper roots and uses root image analysis software to quantify root traits. More recently, X-ray computer tomography (CT) was used for 3D visualization that captures more detailed root architecture traits [97].

Compared to shoots, root data gathering is particularly challenging. This difficulty exponentially increases as roots grow deeper into the ground. Thus, the development of easy-to-use methodologies for root phenotyping remains an important unmet need. The belowground disposition of the root makes it more beneficial to simulate a 3D visualization of the root system architecture (RSA) when compared to SSA. Similarly to SSA, the most common approach in modeling RSA is also the L-system [91,98].

Researchers have long been engaged in constructing virtual models of roots and using those models to simulate the effects of changing environmental conditions on RSA, with the aim of prioritizing interventions that could increase crop yield [99]. The development of root modeling platforms dates back to the 1970s [100,101]. Later, more advanced modeling platforms were created. ROOTMAP [102] is a notable 3D model focusing on fibrous root systems and their proliferation. Other platforms, like RootTyp [103,104] and DigR [105], highlight the influence of soil in root development, while SPACSYS [106] delves into root–soil interactions and their impact on crop yield.

Unlike metabolic models or SSA models, many RSA modeling platforms are not publicly available [107]. This hinders the advancement of the field. The rise of open science spurred initiatives like OpenSimRoot [108], an open-source 3D RSA model accommodating various plant species and focusing on nutrient acquisition feedback to root development. Similarly, CRootBox [109], derived from RootBox [98], fills the gap between in situ and in silico studies by integrating 3D RSA simulations into experimental setups such as cylindrical pots and rhizotrons.

These platforms contribute to untangling the complexities of RSA. More recently, a modeling study [110] proposed an approach for predicting root/shoot system architectures. Their iterative and stochastic method generates elongation length, root direction, and branching until building an individual plant root architecture. Notably, their approach allows for the modular introduction of hormonal and metabolic factors influencing architecture development, predicting their effects on various root traits like length, branching, volume, and surface area.

##### Creating a Root System Architecture

Figure 3 summarizes a three-step workflow for building models for root system architectures (RSAs). In a parallel situation to that of plant shoots, plant roots may be classified either as taproots for dicots or fibrous roots for monocots. Taproots have one primary root (or main root axis) while fibrous roots have several additional root axes, for example, seminal and nodal roots. As such, the initial step in creating a model for an RSA is to decide which type of root one needs to model.

Subsequently, defining a root axis is essential for constructing the RSA model. Figure 3 illustrates a root segment of the RSA, consisting of a main root axis that displays secondary root branching in the form of lateral roots. This main root axis accurately represents a taproot. In contrast, modeling fibrous roots requires the generation of additional axes to describe seminal and nodal roots.

A step-by-step description of such a model-building process can be found in recent literature [110]. That process requires defining at least two parameters: the growth elongation rate and the growth angle, which are important for growth elongation rate and growth angle, respectively. The root system architecture is initialized at the origin in 3D space. Through iterative steps, the approach decides the magnitude of the elongation and direction of the root for each step. Then, the branching points are determined stochastically, from which secondary roots emerge and grow, generating a full architecture. This model outputs a 3D visualization of the RSA of a plant, something hard to imagine experimentally. Once the RSA is generated, various traits, such as root length, lateral root branching, volume, and surface area, can be quantified.

### 2.3. Resource Acquisition

Resource acquisition in plants encompasses obtaining energy and fixing carbon via photosynthesis and collecting minerals and fixing nitrogen through the roots. Modelling plant resource acquisition is often but not always connected to modelling RSA and SSA. Understanding how changes in SSA or RSA influence light acquisition via photosynthesis and water and mineral nutrient uptake from the soil is usually the primary objective of resource acquisition models.

In the context of resource acquisition, researchers use 3D SSA/RSA models to identify factors that change the architectures in ways that maximize a plant’s resource acquisition. Examples include modelling the effect of changing the leaf’s area on the amount of light captured by the plant [111,112,113], or simulating the effect that changing the spread and depth of the root into the soil has on mineral, nitrogen, or water uptake through the root [114].

#### 2.3.1. Photosynthesis

##### Applications

Photosynthesis sustains life on our planet through the supply of food and oxygen [115], allowing plants to grow, develop, and reproduce [116]. This process lets plants capture energy from light, use that energy to fix atmospheric carbon from CO_2_ into sugar, and release oxygen into the air as a byproduct.

Photosynthesis has two phases. The first phase converts energy from light into a chemically usable form. The second phase fixes atmospheric CO_2_. By and large, there are three different types of metabolism to fix that CO_2_: C_3_-, C_4_- or Crassulacean acid metabolism (CAM) [116,117,118].

Figure 4 illustrates how to model photosynthesis. The process initiates with the decision regarding which type of metabolism needs to be modeled (C_3_-, C_4_-,or CAM). Model inputs always include (but are not limited to) light and CO_2_. Regardless of the type of photosynthetic metabolism, three broad approaches can be used to build the model: empirical, mechanistic, or hybrid approaches that combine the first two [116]. In addition, and independent of the approach, the models can be either deterministic or stochastic and simulations can run on continuous or discrete time steps. The study in [116] highlighted that decisions regarding which approaches to use should consider minimizing the number of model parameters and correctly representing physical, chemical, and biological laws. One should also consider minimizing the variance of the simulation outputs and the deviation between predicted model values and measured experimental values.

##### Creating a Photosynthesis Model

The Farquhar–von Caemmerer–Berry (FvCB) model pioneers the mathematical modeling of the photosynthesis of C_3_ plants [117]. The FvCB model calculates the CO_2_ assimilation using Equation (4) [116,117,119]:(4)$$A=V_{C}-0.5V_{O}-R_{d}.$$

Here, $V_{C}$ and $V_{O}$ represent carboxylation rate and oxygenation rate, respectively, and $R_{d}$ is the mitochondrial respiration rate.

C_4_ photosynthesis relies on the coordination between mesophyll and bundle sheath cells within leaves. Initially, CO_2_ is fixed into a C_4_ acid, likely in the mesophyll, by phosphoenolpyruvate carboxylase (PEPC) [118,120]. This 4-carbon compound then moves to the bundle sheath, where it undergoes decarboxylation, providing CO_2_ for Rubisco to initiate the regular C_3_ carbon fixation process. Therefore, modeling C_4_ photosynthesis entails additional initial steps, along with a mathematical model of C_3_ photosynthesis (see [118] for details).

Plants in arid regions, characterized by less than 25 cm of rainfall per year, have adapted to dry environments by evolving CAM metabolism, a type of photosynthesis that fixes CO_2_ more efficiently while reducing evapotranspiration. CAM photosynthesis operates through a circadian metabolic cycle, with stomata opening at night and closing during the day to minimize water loss. During the night, PEPC fixes CO_2_ to produce malic acid, which is stored in the cell vacuole. During the day, the vacuole releases malic acid, which is then decarboxylated through the Calvin Cycle. A recent review [121] presented several mathematical models of CAM photosynthesis, including ODE-based models [122,123,124] and FBA-based models [125].

In photosynthesis models, the typical output variable is the rate at which plants fix atmospheric CO_2_ and/or the rate at which plants use that CO_2_ to produce sugars and other compounds. Various definitions of photosynthetic rate exist in the literature, all linked to the plant’s use of atmospheric CO_2_ to produce sugars [116,117].

#### 2.3.2. Nutrient Uptake Model

##### Applications

One of the primary functions of roots is to acquire water and nutrients from the ground and supply them to meet the metabolic needs of plants. Nutrient acquisition models are normally layered on top of RSA in silico models [108,109,126,127]. These models are highly valuable for investigating the effects of rapidly changing environmental conditions on plant growth, and for prioritizing costly field trials, thereby saving time and money [128]. They are also useful in combination with traditional approaches to develop innovative practices in fertilizer management [129]. In addition, quantifying root nutrient uptake provides insights into potential crop yield.

Over the years, the development and application of several models to quantify root nutrient uptake have been on the rise, aiming to optimize fertilizer usage in agriculture. Initially, nutrient uptake was modeled as directly proportional to the available nutrient concentration in the soil, with root absorption power serving as the constant parameter [130]. However, solving the system of partial differential equations (PDEs), whether analytically or numerically, posed a significant challenge. To address this issue, a study [131] introduced the Crank–Nicolson method to numerically solve their nutrient uptake model. Building upon this approach, another study [132,133] incorporated root system growth into their models using the same numerical method. Later, the roots of rapeseed were modeled to investigate the influence of root hairs on the uptake of phosphorus [134]. These studies were extended [135] to other plant species, which quantified how plant phosphorus uptake strongly influenced the number, length, and radius of root hairs.

The Nye–Tinker–Barber (NTB) model became the most frequently used for modeling plant nutrient uptake [136,137]. A study [138] extended this model, demonstrating the existence of an explicit closed-form solution to the diffusion–adsorption problem [139,140]. Additionally, it was further extended [141] and analytical solutions were provided to convection–diffusion equations applicable to general solute nutrient uptake scenarios.

Recent advancements include the application of perturbation expansion methods to approximate nutrient flux and concentration solutions [142]. This approach divided the rhizosphere into inner and outer fields, aligning with the root surface, and derived approximate analytical solutions for nutrient uptake flux at the root surface and global nutrient diffusion of the NTB model. Another study [143] contributed to finding analytical solutions using Laplace transforms.

##### Creating a Root Nutrient Uptake Model

Figure 4 illustrates the modeling process of nutrient uptake, beginning with the definition of the biological problem. This involves identifying the specific nutrient being modeled, which can be categorized as either mobile (such as nitrogen, sulfur, boron, manganese, and chlorine) or immobile (including phosphorus, potassium, magnesium, calcium, copper, iron, and zinc) in soil. A classification of the mobility of mineral nutrients in soil [144] is presented in Appendix A.

Following the classification of nutrients, diffusion equations mathematically represent the transport process from the soil to the root. These equations are typically described using PDEs, which incorporate both time and space as independent variables [145]:(5)$$\frac{\partial c\left(x,t\right)}{\partial t}=\frac{\partial J\left(x,t\right)}{\partial x},$$
where the concentration (*c*) changes over time (*t*) and the flux (*J*) depends on the spatial variable (*x*).

This simple diffusion equation becomes more complex as additional spatial features of solute transport are incorporated. A review study [146] categorized various solute uptake models from the literature based on their complexity, including models representing diffusion alone, diffusion with advection, diffusion with reaction, and diffusion with both advection and reaction.

Once root nutrient uptake is mathematically represented using PDEs, the next step is to determine their solution, which can be either analytical or numerical. While finding analytical solutions can be challenging, a widely used method is non-dimensionalization, which simplifies PDEs by introducing non-dimensional variables [138]. Non-dimensionalization facilitates the process of finding an analytic solution [147]. However, when analytical solutions are hard to find, numerical simulations are typically sought through simulations, often by discretizing time and space. A study [148] extensively reviewed different numerical methods for discretization.

After determining the solution, various analyses can be performed, including parameter estimation, sensitivity analysis, or the prediction of soil nutrient uptake fluxes and concentrations.

### 2.4. Shoot–Root Interaction

#### 2.4.1. Applications

The allocation of resources in plants is a crucial ability that determines their survival in challenging environmental conditions. Current climate trends emphasize the need for crops that have phenotypic plasticity and can adapt to growing with less water and more heat. This phenotypic plasticity comes from the capacity of a given genotype to express different phenotypes in response to varying environmental conditions [149]. Modeling shoot–root interactions is especially useful for simulating different scenarios and predicting the range of the phenotypic plasticity of plants in different surroundings. In addition, modeling shoot–root dynamics allows for a further exploration of plant productivity, considering the allocation of nutrient minerals from uptake and carbon from photosynthesis. This integration creates a multiscale mathematical model capable of facilitating in silico experiments ranging from the metabolic level to the whole plant level.

Researchers proposed many transport-resistance (TR) models to analyze plant growth dynamics. A TR model that pioneered this field [150] introduced a leaf–stem–root model that accounted for the utilization and transportation of photosynthate among different parts of the plant. In a subsequent study [151], they constructed a shoot–root model focusing on the exchange of photosynthetic output between the shoot (carbon) and root (nitrogen) uptake. Additionally, he incorporated the chemical and biochemical conversion of these substrates into structural dry matter [152]. This was expanded [153] by considering the effect of water potential and transpiration in his model. He utilized the Münch flow mechanism of phloem translocation and introduced a switch based on concentration gradients. In a more recent study on phosphorus dynamics [154], researchers combined experiments and models to investigate shoot–root interaction over a day-and-night cycle, using *Petunia hybrida* as a model organism.

There are other whole plant-level models that consider root–shoot interaction and are not TR models. For example, CPlantBox [155] and PiafMunch [127,156] applied cohesion-tension theory for water flow and Münch flow theory for carbon flow.

#### 2.4.2. Creating a Shoot-Interaction Model

Figure 5 outlines the modeling process for shoot–root interactions, depicting the input compartment consisting of resources acquired from photosynthesis and nutrient minerals from root uptake. These models primarily focus on resource allocation, emphasizing the exchange and transfer of resources between the shoot and root. In describing the utilization and transportation of substrates, most models employ the TR approach. This modeling approach is favored for its realistic representation of transport, accounting for substrate losses during transport and translating C/N fixation into root growth, using functions that convert volume into plant structure [157].

Still, different models use alternative strategies. For example, the teleonomic model [151] prioritizes factors like the shoot–root ratio, growth, and substrate concentrations based on environmental conditions [158]. Other models give more importance to other factors, such as soil water potential and root temperature effects on the plant partitioning of photosynthate [159].

Regardless of the approach used, shoot–root models need to account for the interactions between two interconnected compartments: the shoot system and the root system [154]. The shoot part may range from simple shoot concentrations of substrates to more complex models specifying shoot concentrations of stored starch and soluble sugars, including shoot structure/architecture and mineral nutrients from the roots such as nitrogen, phosphorus, and potassium [127,155,160]. Similarly, the root part comprises exchanged substrate concentrations from the shoot, root structure/architecture, and nutrient uptake from the soil [154,155]. The exchange of soluble sugars from the shoot and mineral nutrients from the root is driven by a transport function [152], with the growth rate dependent on the substrate concentrations it contains.

These models often output the concentration of different substrates for shoot and root over time, mathematically represented by ordinary differential equations (ODEs). Thus, they enable in silico stability and sensitivity analysis of the shoot–root ratio under various environmental conditions.

## 3. Discussion

### 3.1. The Potential of Multiscale Mathematical Modeling

Plant synthetic biology aims to engineer plant biological systems to improve crops through the direct manipulation of genetic circuits in plant genomes [10]. Given the complexities of such manipulations, they come associated with potentially serious biosafety risks [19,161], ethical issues [20], and economic risks [23]. These risks are interconnected, with ethical issues serving as the broader umbrella. Multiscale mathematical models can be a valuable tool for risk mitigation through the in silico simulation and analysis of those effects. These models provide another layer of quality control for the design stage of the design–build–test–learn (DBTL) cycle of synthetic biology [162]. Ultimately, this approach can expedite the development of new crop varieties and reduce the costs associated with that development [163].

While crucial to test the effect of interventions, traditional plant breeding techniques and experimental studies are time-consuming and unpredictable [74]. We can use multiscale mathematical models to predict how genotype modifications affect plant’s phenotypic traits. Integrating in silico investigation through multiscale mathematical models using the DBTL cycle of synthetic biology enhances the speed and accuracy of predictions made about the effect of genome manipulation on in planta implementation [163,164]. In addition, we can fine-tune their precision in quantifying the changes caused by the genetic intervention to molecular, architectural, resource, and whole-plant levels by incorporating experimental information that becomes available during the model-building process.

Understanding and correctly representing the interplay between those levels is crucial for improving plant growth and productivity and optimizing fertilizer management while reducing environmental impacts. For instance, growing shoots and roots are responsible for resource acquisition, which is then used to produce sugar that is consumed for the shoot and the root to continue to grow. Several studies [165,166,167] pinpoint the influence of carbon, nitrate, and sugar in shoot and root development. This and many other nonlinear internal feedback loops naturally exist in biological systems [60,168]. Accurately predicting the effect of those regulatory loops on the dynamic behavior of the plant requires using multiscale mathematical models. The predictions can subsequently be validated by comparing them to experimental evidence [31]. A review [169] discusses several designs of synthetic gene circuits for plants, which can be modeled using a multiscale mathematical model approach [31], following the methodology presented in Figure 1.

Accounting for the complexity of plant systems requires multiscale mathematical models, as single-scale models cannot capture the intricate dynamics of the whole system [170]. Figure 1 illustrates an iterative process for creating these models, allowing entry at any step of the model-building process. This approach helps in understanding how molecular changes propagate throughout the organism, informing strategies for genome modification to achieve specific objectives.

Despite the beneficial contribution of this approach to the advancement of plant synthetic biology, there remains a significant gap between scales in terms of the available knowledge and methodological tools for creating and analyzing models at each scale. This is evident when comparing, for example, metabolic-level models to shoot/root system models. Such gaps complicate the integration between scales within a model. This integration is crucial for evaluating how genetic manipulations interact with environmental and metabolic cues, influencing plant development. The modular nature of plant systems suggests an approach to facilitate scale integration in multilevel models: make the models representing the different levels as modules [30,31,171]. Also, there is a continuous effort to build Crops In Silico [170], a platform for plant mathematical models that highlights the value of a multiscale approach.

A critical aspect of this process is describing the connections between scales or translating the output of one model into input for another. One solution is to represent each scale in a separate sub-model, using appropriate mathematical representations and tools, and interconnect them through relevant variables. This approach has been successfully applied to create multiscale mathematical models of several microorganisms [27,28,29,30].

The development of useful multiscale plant models depends on accurate parameter measurements. Advanced plant phenotyping technologies [95,96,97] are leading to an era of big data, providing modelers with highly accurate measurements. A study [172] proposed a modeling workflow centered on digital twins, using X-ray computed tomography (CT) to capture detailed root images for constructing root skeletons and RSA models. This approach allows for a comprehensive analysis of plant phenotyping data, providing deeper insights into various plant functions [173].

For example, ongoing efforts aim to optimize maize production of a hormone called strigolactone (SL), to develop maize strains resistant to drought and pests, specifically *Striga*. SLs are known to influence root development and act as germination stimulants of *Striga*, with maize primarily producing strigol-type SLs. Initial modeling efforts [60] quantified SL production and identified key enzymes to target not only for modulating its production but also to shift the predominant SL type from strigol to orobanchol. Such effort could result in maize crops that are less susceptible to *Striga* infestation. Further studies [110] modeled the effects of varying SL concentrations on RSA, demonstrating changes in root elongation and lateral root branching density. These changes determine the number of additional SLs required to achieve longer and denser roots. In relation to the workflow, modeling root water uptake could reveal how strigolactone modification enhances maize resistance to drought.

The construction of whole-plant digital twins is likely to be forthcoming. Building whole-plant digital twins will likely be facilitated by artificial intelligence (AI). AI algorithms can analyze experimental data to inform model-building choices and later analyze data generated from digital twins, offering insights into plant growth, development, and responses to environmental stimuli. Leveraging digital twin technology alongside AI has the potential to optimize agricultural processes, improve crop yields, and contribute to sustainable food production. This holistic approach enhances our ability to analyze complex plant systems and paves the way for transformative developments in agriculture, helping with global food security challenges.

### 3.2. Limitations of Multiscale Mathematical Modeling

Modeling complex plant systems requires numerous parameter values, which can constrain the predictive power of models. Technological limitations often make it challenging to obtain certain parameter values, particularly at the molecular level. Detailed multiscale mathematical models also require substantial computational resources, leading to time constraints. As technology evolves, overcoming these limitations will be pivotal for realizing the full potential of multiscale mathematical modeling in plant research.

The integration of different scales within a multiscale framework is essential but challenging. The generalization of sub-models is possible but requires adaptation of the models in ways that depend on the study’s specific objectives. For instance, the whole-cell model [27] offers intricate detail compared to a strigolactone biosynthetic model [60], reflecting distinct research goals.

A significant barrier in plant modeling is the disconnection between wet lab and dry lab researchers. Experimental findings often do not provide the necessary parameter values for mathematical models. Bridging this gap requires collaboration between researchers from both domains, facilitating data-driven model-building and model-assisted experiments. This interdisciplinary approach is crucial for advancing plant modeling.

### 3.3. Concluding Remarks

Multiscale mathematical modeling holds immense potential for characterizing biological phenomena and enabling targeted bioengineering for improved crop productivity and sustainability. However, integrating single-scale models into a comprehensive multiscale framework remains a critical challenge. Exploring diverse modeling approaches at each scale and harmonizing them into a unified whole-plant model is essential.

Deciphering and integrating different single-scale models are ongoing challenges in constructing a multiscale mathematical model. This review proposes a framework for constructing individual sub-models tailored to plants within plant synthetic biology. The application of multiscale mathematical modeling in plant biology is a fertile area for future exploration. It has already proven invaluable in enhancing our understanding of plant complexity and driving advancements in agricultural research. Encouraging more applications of multiscale mathematical modeling in plants can contribute to productive and sustainable agriculture.

In summary, integrating multiscale mathematical modeling approaches promises to revolutionize our understanding of plant biology and enhance agricultural productivity. By leveraging diverse modeling techniques and frameworks, researchers can gain deeper insights into the intricate dynamics of plant systems, leading to more informed decision-making and innovative solutions for global food security challenges. Realizing this potential will require interdisciplinary collaboration, technological advancements, and the ongoing refinement of modeling methodologies to effectively address the complexities of plant biology.

## Figures and Tables

**Figure 1 plants-14-00470-f001:**
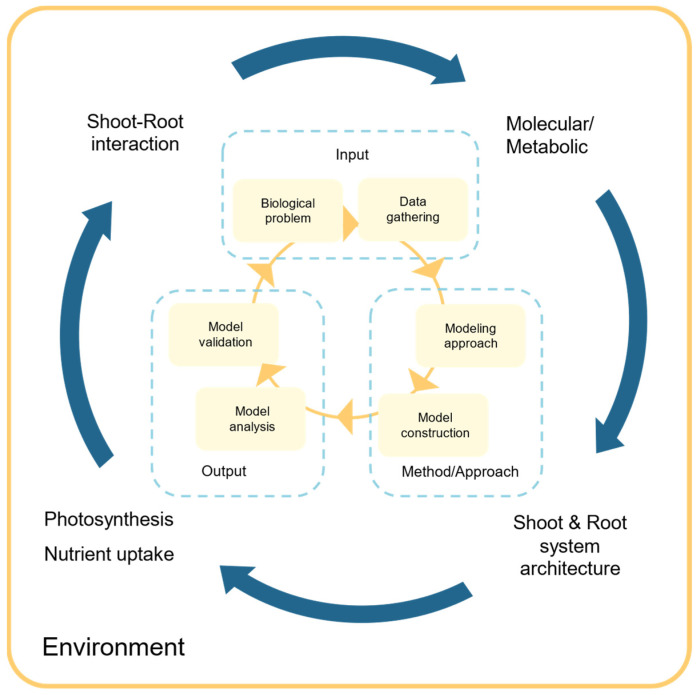
Circular workflow for minimalistic multiscale mathematical modeling. The outer circle proposes the model’s levels and their potential interactions. The inner circle shows the mathematical modeling methodology. The modeling workflow was divided into three compartments: input, method/approach, and output. The outer circle shows the different scale in plant modeling. The inner circle procedure is applied to each scale in the outer circle.

**Figure 2 plants-14-00470-f002:**
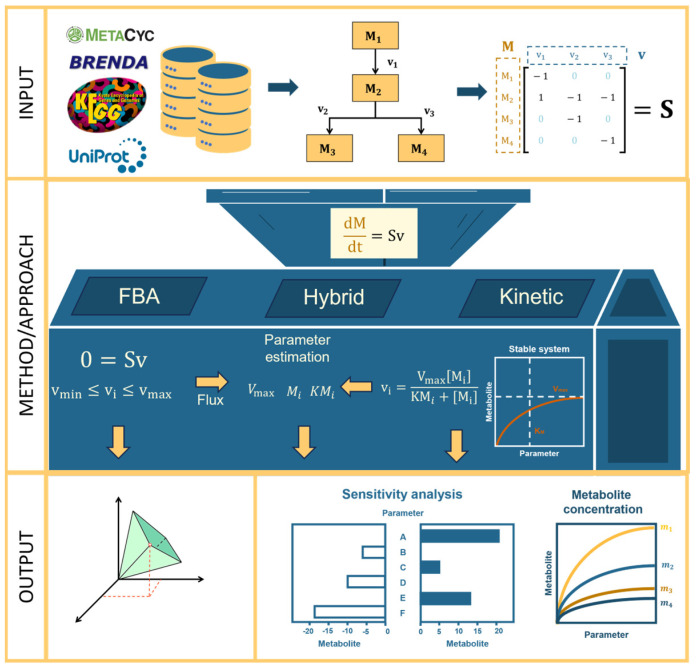
A proposed workflow for modelling metabolism. The workflow is subdivided into three compartments: input, method/approach, and output. The input illustrates necessary a priori details for building a metabolic model. These details include data collection, pathway reconstruction, and stoichiometric matrix calculation. In the second step, the method/approach compartment illustrates the process for FBA, kinetic, and hybrid model building. Finally, the output compartment highlights the possible results from each of the different approaches, together with types of analysis that can be performed. The arrows represent the flow of steps in the process.

**Figure 3 plants-14-00470-f003:**
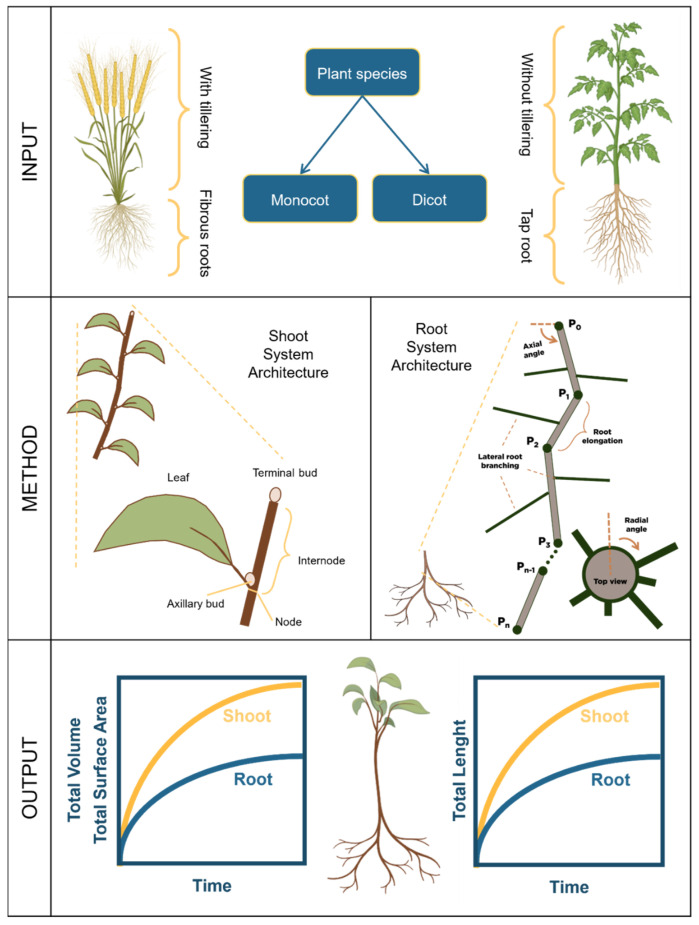
Proposed workflow for modeling shoot and root system architectures. The input panel includes determining whether the plant species to model is a monocot or dicot. The method panel presents the basic modeling unit, which is either stacked up or iterated to form the full architecture. The output panel gives the quantified traits of the shoot and root architecture.

**Figure 4 plants-14-00470-f004:**
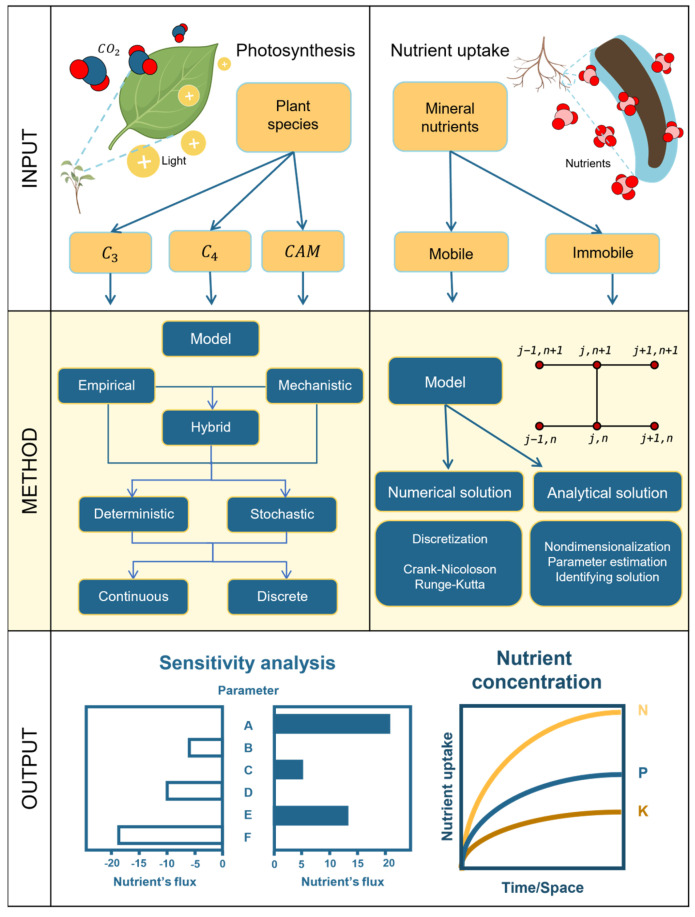
Proposed workflow for modeling resource acquisition. Shoot photosynthesis and root nutrient uptake models need to be integrated. The input panel splits into acquisition of light and CO_2_ in the shoot and acquisition of mineral nutrients in the root. The methods panel summarizes available methods to build each type of model. The output panel illustrates shoot and root resource acquisition traits that can be quantified using the models.

**Figure 5 plants-14-00470-f005:**
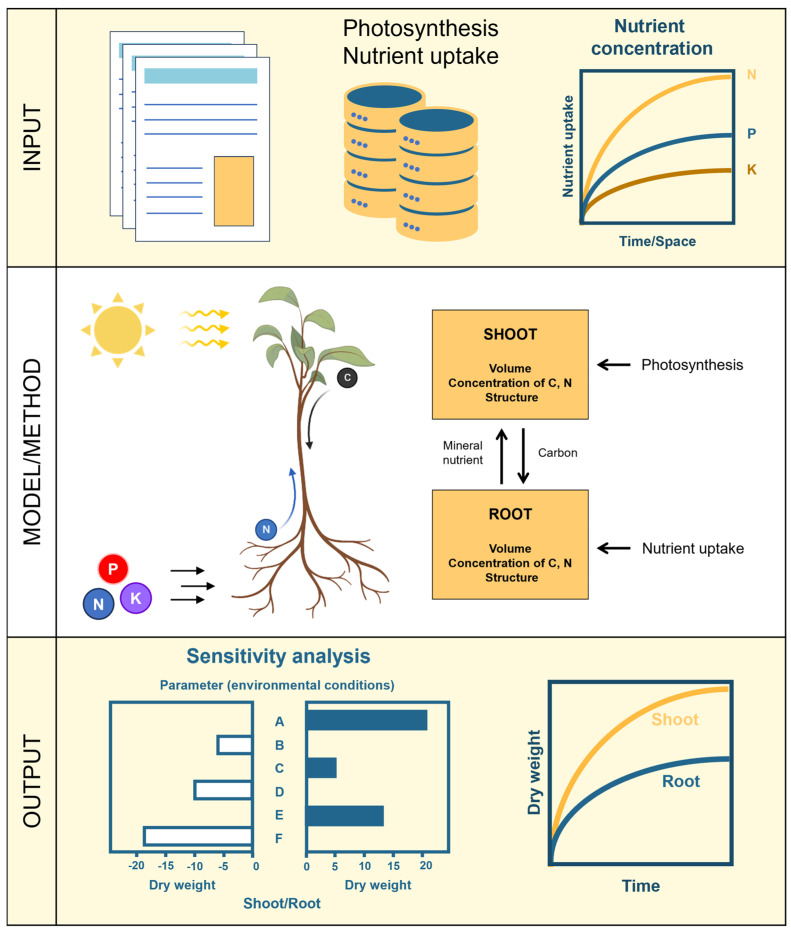
Proposed workflow for modeling shoot–root interactions.

## Data Availability

No new data was generated during this research.

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
