# Peer review of "Multiscale Mathematical Modeling in Systems Biology: A Framework to Boost Plant Synthetic Biology"

_plants, 2025, doi:10.3390/plants14030470_

Round 1

Reviewer 1 Report

Comments and Suggestions for Authors

The manuscript aim to give a comprehensive review on multiscale mathematical modeling in plant synthetic biology. But their only reviewed some computational system biology modeling in plant science, with a special focus on metabolic and  shoot-root interaction, no contents are related plant synthetic biology. My opinion is rejection.

Comments on the Quality of English Language

The manuscript aim to give a comprehensive review on multiscale mathematical modeling in plant synthetic biology. But their only reviewed some computational system biology modeling in plant science, with a special focus on metabolic and  shoot-root interaction, no contents are related plant synthetic biology. My opinion is rejection.

Author Response

Comment 1: The manuscript aim to give a comprehensive review on multiscale mathematical modeling in plant synthetic biology. But their only reviewed some computational system biology modeling in plant science, with a special focus on metabolic and  shoot-root interaction, no contents are related plant synthetic biology.

Reply 1: We acknowledge your concern and would like to clarify our intentions and focus. While the manuscript emphasizes multiscale mathematical modeling in plant sciences, including computational system biology approaches, our aim was to highlight the foundational frameworks and methodologies that underpin plant synthetic biology rather than provide an exhaustive review of plant synthetic biology-specific case studies. By focusing on metabolic and shoot-root interactions, we sought to address core modeling principles that are directly applicable to synthetic biology efforts, such as designing and predicting synthetic metabolic pathways and engineering robust plant systems. 

Reviewer 2 Report

Comments and Suggestions for Authors

In this manuscript, the authors review the multiscale mathematical modeling in plant synthetic biology. The authors emphasize the need for interdisciplinary collaboration and technological advancements to use these models for sustainable agriculture and global food security.

I have the following comments:

- Consider simplifying the technical sections for readability and approaching scientists with different backgrounds. One example if from lines 201-201, where "mathematical formalisms" were mentioned. Instead, it could be mentioned that Equation 1 represents the change in metabolite concentrations over time. This requires selecting a mathematical representation, etc.

- Specific examples to practical applications could be given in some sections, for example lines 240-243 a general statement is given, could this be mentioned for a specific crop and application?

-Is it possible to expand or provide definitions for the specific photosynthesis terms in lines 371-374? Like including what makes C3, C4 and CAM different.

Author Response

Comment 1:  Consider simplifying the technical sections for readability and approaching scientists with different backgrounds. One example if from lines 201-201, where "mathematical formalisms" were mentioned. Instead, it could be mentioned that Equation 1 represents the change in metabolite concentrations over time. This requires selecting a mathematical representation, etc.

Reply 1: 

Thank you for this comment. In previous line 201 and now lines 212-216, we simplified the language and use “mathematical representation” instead of “mathematical formalism.”

“To construct a kinetic model, the step begins with the mathematical representation of the change in metabolite concentrations over time in Equation (1). Then select suitable mathematical representation to describe the kinetics of individual processes and their dependence on substances directly influencing flux. Various representations, including linear, rational, and power laws, among others, are available [60].”

Comment 2: Specific examples to practical applications could be given in some sections, for example lines 240-243 a general statement is given, could this be mentioned for a specific crop and application?

Reply 2: 

We added Table S2 to the supplementary materials, which listed available models across different plant species. The sentences in lines 240-243 are now lines 251-257 which states the following:

“Developing a 3D architecture visualization model is crucial to predict plant shoot/root growth and development, facilitating reasonable approximate estimations of crop productivity [68,69]. Thus, there has been a growing interest in building mathematical models of the shoot and root system architecture to do in silico investigation in plants, some of these modeling efforts are listed in Table S2. However, this field is still less mature than that of metabolic modeling and methodological development is ongoing.”

Comment 3: Is it possible to expand or provide definitions for the specific photosynthesis terms in lines 371-374? Like including what makes C3, C4 and CAM different.

Reply 3: We further specified this in the succeeding paragraphs in lines 406-417, where we also highlight their different processes.

“C4 photosynthesis relies on the coordination between mesophyll and bundle sheath cells within leaves. Initially, CO2 is fixed into a C4 acid, likely in the mesophyll, by phosphoenolpyruvate carboxylase (PEPC) [118,120]. This 4-carbon compound then moves to the bundle sheath, where it undergoes decarboxylation, providing CO2 for Rubisco to initiate the regular C3 carbon fixation process. Therefore, modeling C4 photosynthesis entails additional initial steps, along with a mathematical model of C3 photosynthesis (see [118] for details).”

Reviewer 3 Report

Comments and Suggestions for Authors

Dear Authors,

The manuscript addresses major current issues such as climate change, agricultural losses and the need for sustainability, presenting synthetic biology and mathematical modeling as innovative solutions. The proposal of using multiscale mathematical models (from molecular to ecosystem level) is valuable to understand and optimize the complex interactions of plants.

Please consider the following points for a maximum impact of the manuscript:

While the document proposes solutions to mitigate biosecurity risks, it does not fully explore the ethical issues or economic risks associated with the large-scale adoption of synthetic biology. Can you identify some of the possible risks? It is very important to know where we are heading.

Are there any examples of large-scale implementation of a mathematical model in the domain of life sciences? 

Have you thought about a detailed strategy for bridging the gap between mathematical theory and practical applications?

Specific comments:

Line 85: One critical aspect of scientific manuscripts is that they are typically redacted in the past tense and in the third person. You should retain from employing personal pronouns such as "I" or "we." Kindly check the whole article and rephrase (lines 85, 86, 97, 99, 105, 110, 111 etc.).

Line: 185: This sentence has no sense! What do you mean? "Let us now look at the modeling process in more detail". Please rephrase!

Lines 338-352: Kindly align the paragraph according to the Justify alignment.

Line 377: Co2 has to be non italic.

Concluding Remarks: Citations are unnecessary in this section of the article, as they are your own interpretations, findings, and explanations. The conclusions are derived from the research that you have conducted and have presented in the manuscript. Consequently, citations are redundant.

Good luck forward in your research.

Author Response

Comment 1: While the document proposes solutions to mitigate biosecurity risks, it does not fully explore the ethical issues or economic risks associated with the large-scale adoption of synthetic biology. Can you identify some of the possible risks? It is very important to know where we are heading.

Reply 1: 

We appreciate the comment of the researcher, with that we added a paragraph in the introduction to talk more about economic and ethical challenges in synthetic biology. This paragraph is in lines 74-83.

“Aside from biological and environmental risks, synthetic biology also presents economic and ethical challenges. Economically, a study on the economics of synthetic biology [23] highlighted the potential “winner-take-all” scenario, leading to monopoly of a company. Still, preserving the balance between open-source platforms and patent protections offers a promising approach to keep the risk manageable. Ethically, challenges include ensuring fair and intergenerational accessibility and maintaining public trust. Existing ethical governance frameworks [20,24,25] provide guidelines for addressing these issues. In relation to in silico investigations, the development of open-source, user-friendly platforms capable of predictive simulation could play a key role in communicating the risk and benefit of synthetic biology to the public.”

Comment 2:  Are there any examples of large-scale implementation of a mathematical model in the domain of life sciences?  Have you thought about a detailed strategy for bridging the gap between mathematical theory and practical applications?

Reply 2: 

One of the strategies that could bridge the gap between mathematical theory and practical applications is digital twin implementation. This involves three layers, a plant experiment, a high-throughput phenotyping, and a mathematical model. We briefly mentioned it in lines 614-622.

“The construction of whole-plant digital twins is likely to be forthcoming. Building whole-plant digital twins will likely be facilitated by artificial intelligence (AI). AI algorithms can analyze experimental data to inform model-building choices and later analyze data generated from digital twins, offering insights into plant growth, development, and responses to environmental stimuli. Leveraging digital twin technology alongside AI has the potential to optimize agricultural processes, improve crop yields, and contribute to sustainable food production. This holistic approach enhances our ability to analyze complex plant systems and paves the way for transformative developments in agriculture, addressing global food security challenges.”

Comment 3:  Line 85: One critical aspect of scientific manuscripts is that they are typically redacted in the past tense and in the third person. You should retain from employing personal pronouns such as "I" or "we." Kindly check the whole article and rephrase (lines 85, 86, 97, 99, 105, 110, 111 etc.).

Reply 3: Thank you for this comment.I agree to the following comments and rewrite the following sentences as followed: Lines 85-86 are now lines 97-100

“Alternatives for integrating models across scales were also explored, with a discussion of the advantages and shortcomings of each approach. Finally, the review provided insights into ongoing efforts to develop an integrated multiscale mathematical model for maize, engineered through synthetic biology to enhance resilience against Striga and droughts.”

We did the same thing to other sentences in first-person, you can find this in the highlighted version.

Comment 4:  Line: 185: This sentence has no sense! What do you mean? "Let us now look at the modeling process in more detail". Please rephrase!

Reply 4: Now, line 195 was improved as suggested and now written as:“The modeling process is examined in greater detail below.”

Comment 5: Lines 338-352: Kindly align the paragraph according to the Justify alignment.

Reply 5: Thank you for this comment. The previous lines 338-352 are now justified and in lines 344-364.

Comment 6: Line 377: Co2 has to be non italic.

Reply 6: We corrected the CO2 in the previous line 377 which is now line 389.

Comment 7: 

Concluding Remarks: Citations are unnecessary in this section of the article, as they are your own interpretations, findings, and explanations. The conclusions are derived from the research that you have conducted and have presented in the manuscript. Consequently, citations are redundant. Good luck forward in your research.

Reply 7: We agreed in the suggestion of the reviewer in removing the citation in the concluding remarks.

Reviewer 4 Report

Comments and Suggestions for Authors

This is a very interesting and promising review of plant synthetic biology,  including the complexities and risks associated.

Authors advocate for the integration of technological advancements in agricultural data analysis to foster a comprehensive understanding of crops across scales, such as metabolic modeling (FBA, kinetic models, hybrid approaches), shoot and root system architectures, and resource acquisition models (e.g., photosynthesis and nutrient uptake), etc.

Authors have described the  utilization of a  bunch of  multiscale mathematical models
but there are certain parts of the main text that seem confusing to me.

For example:

Line 135  model creation and analysis. Table 1 provides a sample of databases containing kinetic … or  …

Line  206 …We can obtain initial parameter estimates for example from databases listed in Table 1 o

 (I cannot find the table 1) .

Line 272. In Table 2, we listed several functional-structural plant modeling (FSPM) platforms

(I cannot find the table 2).

Line 458. A classification of the mobility of mineral nutrients in soil [141] is presented in Table 3.

(I cannot find the table 3).

Line  23 . To illustrate the potential of the combination, we outline

ongoing efforts to develop an integrated multiscale mathematical model for maize (Zea mays L.),

engineered through synthetic biology to enhance resilience against Striga and drought.

This example is mentioned in lines 88-91, and is further explored within the framework of integrated models in the discussion section (lines 493-505).

But it seems to me that based on what was mentioned in the abstract, it is necessary to go deeper.

The figures and the rest of the text are OK.

Minor corrections:

Line 787: "Changing Biosynthesis of Terpenoid Percursors Precursors in Rice..."

Author Response

Comment 1: Authors have described the  utilization of a  bunch of  multiscale mathematical models 
but there are certain parts of the main text that seem confusing to me. 

For example:

Line 135  model creation and analysis. Table 1 provides a sample of databases containing kinetic … or  …

Line  206 …We can obtain initial parameter estimates for example from databases listed in Table 1 o

 (I cannot find the table 1) .

Line 272. In Table 2, we listed several functional-structural plant modeling (FSPM) platforms

(I cannot find the table 2).

Line 458. A classification of the mobility of mineral nutrients in soil [141] is presented in Table 3.

(I cannot find the table 3).

Reply 1: We included Tables 1-3 as supplementary materials as Tables S1-S3, respectively, and corrected the text accordingly.

Comment 2:  Line  23 . To illustrate the potential of the combination, we outline ongoing efforts to develop an integrated multiscale mathematical model for maize (Zea mays L.), engineered through synthetic biology to enhance resilience against Striga and drought. This example is mentioned in lines 88-91, and is further explored within the framework of integrated models in the discussion section (lines 493-505). But it seems to me that based on what was mentioned in the abstract, it is necessary to go deeper. The figures and the rest of the text are OK.

Reply 2: We believe that the reviewer refers to lines 593-605 of the discussion section. We added more details about it as follows which you can also find in lines 618-629:

Comment 3: 

Minor corrections:

Line 787: "Changing Biosynthesis of Terpenoid Percursors Precursors in Rice..."

Reply 3: The original article has this title. We corrected it in our bibliography.

“For example, ongoing efforts aim to optimize maize production of a hormone called strigolactones (SLs), to develop maize strain resistant to drought and pests, spe-cifically Striga. SLs are known to influence root development and act as germination stimulants of Striga, with maize primarily producing strigol-type SLs. Initial modeling effort [57] quantified SL production and identify key enzymes to target not only for modulating its production but also to shift the predominant SL type from strigol to orobanchol. Such effort could results in maize crops that are less susceptible to Striga infestation. Further studies [107], modeled of the effects of varying SL concentrations on RSA, demonstrating changes in root elongation and lateral root branching density. These changes determine the amount of additional SLs required to achieve longer and denser roots. In relation to the workflow, modeling root water uptake could reveal how strigolactones modification enhances maize resistance to drought.”

Round 2

Reviewer 1 Report

Comments and Suggestions for Authors

it's still a  computational system biology modeling review in plant science.

Author Response

Comment 1: "It's still a  computational system biology modeling review in plant science."

Reply 1:   We agree with the reviewer in that we are reviewing computational systems biology methods. Still, we are doing so in the context of their application to Plant Synthetic Biology. To make both of these aspects clear we have change the title of the review to "Multiscale mathematical modeling in systems biology: A framework to boost plant synthetic biology". We have also made small text adjustments throughout the review in order to adjusted to the title change.  

Reviewer 3 Report

Comments and Suggestions for Authors

I am glad to see that the manuscript has been improved according to the recommendations of the reviewers. However, small editing retouches are still needed.

Author Response

Comment 1: "I am glad to see that the manuscript has been improved according to the recommendations of the reviewers. However, small editing retouches are still needed."

Reply 1: We went through the text, further adjusting and polishing it to improve readability. This can be seen in the marked revised version

Reviewer 4 Report

Comments and Suggestions for Authors

The authors have responded to my requests.

Author Response

Comments 1: "The authors have responded to my requests."

Reply 1: thank you. Still, we mad further improvements to the text.